# Evaluation of the Prognostic Impact of SP263-Evaluated PD-L1 Expression in Patients with Stage III Non-Small Cell Lung Cancer (NSLC) Treated with Radio-Chemotherapy

**DOI:** 10.3390/biomedicines12030688

**Published:** 2024-03-19

**Authors:** Jan Nicolai Wagner, Julia Roeper, Lukas Heukamp, Markus Falk, Kay Willborn, Frank Griesinger

**Affiliations:** 1Faculty VI—Medicine and Health Sciences, European Medical School, University Oldenburg, Ammerländer Heerstraße 114-118, 26129 Oldenburg, Germany; julia.roeper@uni-oldenburg.de; 2Hematopathology Hamburg, Fangdieckstraße 75A, 22547 Hamburg, Germany; heukamp@hp-hamburg.de (L.H.); markus.falk@hp-hamburg.de (M.F.); 3Department of Radiotherapy and Radio-Oncology, University Department of Medical Radiation Physics, Pius-Hospital Oldenburg, Georgstraße 12, 26121 Oldenburg, Germany; kay.willborn@pius-hospital.de; 4Department of Hematology and Oncology, University Department of Internal Medicine-Oncology, Pius-Hospital Oldenburg, Georgstraße 12, 26121 Oldenburg, Germany; frank.griesinger@pius-hospital.de

**Keywords:** local advanced NSCLC, stage III, PD-L1, radio-chemotherapy, lung cancer

## Abstract

Background: The PACIFIC study showed that after radio-chemotherapy, patients with NSCLC derived a benefit in PFS and OS when treated with durvalumab. This effect was limited to patients with a PD-L1 expression of >1%, partly because the outcome in the observational control arm was surprisingly favorable. Thus, it could be speculated that a lack of PD-L1 expression confers a favorable outcome for patients with stage III NSCLC. Methods: Clinical data, PD-L1 expression, predictive blood markers, and the outcomes of 99 homogeneously treated patients with stage III NSCLC were retrospectively captured. Statistical analyses using the log rank test were performed. Results: The median OS of patients with an expression of PD-L1 < 1% was 20 months (CI 10.5–29.5) and the median OS of patients with an expression of PD-L1 ≥ 1% was 28 months (CI 16.5–39.2) (*p* = 0.734). The median PFS of patients with an expression of PD-L1 < 1% was 9 months (CI 6.3–11.6) and the median PFS of patients with an expression of PD-L1 ≥ 1% was 12 months (CI 9.8–14.2) (*p* = 0.112). Conclusions: The assumption that the lack of PD-L1 expression represents a favorable prognostic factor after radio-chemotherapy vs. PD-L1 expression > 1% was not confirmed.

## 1. Introduction

The therapy of NSCLC has been revolutionized using immune checkpoint inhibitors (ICI). They are now established in the metastatic setting as a first-line treatment, as a single agent or in combination with platin-based chemotherapy, and have conferred both increased response rates, progression-free survival (PFS), and overall survival (OS) compared to combination chemotherapy alone [1,2,3,4]. In the metastatic setting, PD-L1 has been established as a predictive factor for response, PFS, and OS on ICI and is the only biomarker for the stratification of ICI therapies, mainly due to licensing reasons [5]. Data on PD-L1 expression regarding its prognostic impact are sparse. When comparing the control arms of the licensing studies such as Keynote 189, Impower 150, and Keynote 407 for the different PD-L1 expression subgroups (<1%, ≥1–49%, ≥50%), no obvious differences in the outcome can be discerned, suggesting that in the metastatic setting, the prognostic impact of PD-L1 expression is most likely limited [6,7,8]. In the PACIFIC study, patients with stage III NSCLC who were not eligible for surgery were included and were treated with radio-chemotherapy with a radiotherapy of at least 54 Gray (Gy) [9]. Patients who did not progress after radio-chemotherapy and who did not show any signs of radiation-induced pneumonitis were randomized to receive durvalumab vs. placebo. A statistically significant benefit for PFS and OS was shown for patients receiving durvalumab in the ITT population. The updated 5-year OS for patients receiving durvalumab was 47.5 months (38.4–52.6) versus 29.1 (22.1–35.1) for patients receiving placebo [10]. The OS HR was 0.71. In a post hoc analysis, the benefit for OS was statistically significant in the group with a PD-L1 expression of ≥1%, but not in the group with an expression of <1%. One explanation why the PD-L1 < 1% group might not derive a benefit from durvalumab consolidation could be that the control arm of this cohort was significantly better than the control arm of the PD-L1 ≥ 1% group. Therefore, the hypothesis was generated that a lack of PD-L1 expression could represent a favorable prognostic factor in NSCLC after radio-chemotherapy. To test this approval-relevant hypothesis, a retrospective cohort study of patients treated with radio-chemotherapy, similar to the PACIFIC trial and in whom PD-L1 expression was known, was performed. The outcome (PFS and OS) was analyzed according to PD-L1 status (≥1% vs. <1%). Additionally, the prognostic impact of PD-L1 expression was investigated for different cut-off values.

The study situation is limited. The study by K. Gennen et al. included 31 patients. They found that PD-L1 expression < 1% on tumor cells was associated with improved OS, PFS, and local control in patients treated with concurrent CRT [11]. In contrast to these results, the study by T. Tokito et al. with 74 patients with stage III NSCLC showed that PD-L1 expression was not correlated with PFS and OS [12]. This has been confirmed by the study of A. Tufman et al. In their study, 78 patients with stage III NSCLC were included, and PD-L1 expression did not correlate with PFS following RTCT [13]. A study from Vrankar et al. with 117 patients confirmed these results as well [14]. With 99 patients, our study is the second largest analyzing the prognostic impact of PD-L1 expression after radio-chemotherapy and the one with the most accurate staging by using PET-CT in all patients. The results are shown in Table 1.

Additional potentially prognostic serum markers (CRP and albumin) were evaluated. In a Chinese cohort, these parameters were deemed prognostically significant for PFS and OS in a population of patients, including all stages [15]. Therefore, our cohort was used to evaluate the prognostic significance of CRP and albumin after radio-chemotherapy for stage III NSCLC patients.

## 2. Materials and Methods

To match the criteria of the PACIFIC study, eligible patients for this retrospective analysis were primary case patients being treated at the lung cancer center at Pius Hospital, Oldenburg, who had stage III NSCLC histologically or cytologically diagnosed by staging procedures including PET-CT (Siemens Biograph 6, Siemens Helathineers, Forchheim, Germany)) and MRI (3T MRI Magnetom Verio (Siemens Helathineers, Forchheim, Germany)) of the brain. This study is a complete survey of real-world data from daily routine care patients who were treated at the clinic during this period and met the inclusion criteria. Staging was performed according to the Staging Manual in Thoracic Oncology, version 8, of the International Association for the Study of Lung Cancer. The patients had received two or more cycles (defined according to local practice) of platinum-based chemotherapy (containing etoposide, vinblastine, vinorelbine, a taxane, or pemetrexed) concurrently with definitive radiation therapy (54 to 66 Gy). The mean dose to the lung was less than 20 Gy; the V20 (the volume of lung parenchyma that received 20 Gy or more) was less than 35%. Additional inclusion criteria were no disease progression, no pneumonitis after radio-chemotherapy, and an age of 18 years or older. Patients who did not sign the patient informed consent form of the Pius Hospital were not included. Clinical characteristics captured included age, gender, histology, smoking status, ECOG, and type of first-line treatment. Smoking status was defined as follows: ex-heavy smokers: more than 10 pack years and having quit smoking for more than 10 years; current heavy smokers: current smokers with more than 10 PY; light smokers: less than 10 PY and having quit more than 10 years; never smokers: less than 100 cigarettes in a lifetime.

### 2.1. Study Design and Treatments

This retrospective study included 134 patients who were treated at the Pius Hospital from 2011 to 2018. The patients were selected from the lung cancer database of the Pius Hospital. In the end, 99 patients were included in the analysis; 35 patients could not be evaluated retrospectively for PD-L1 expression as not enough tumor material was left. The evaluation of PD-L1 expression on tumor cells was carried out with the anti-PD-L1 antibody SP263 on the Ventana platform at the Hematopathology Hamburg, to mimic the methods of the PACIFIC study. CRP and albumin values were used if they were analyzed no more than 14 days before the start of radio-chemotherapy.

### 2.2. End Points and Assessments

The primary end points were progression-free survival and overall survival. PFS was defined as the first day after the last radio-chemotherapy treatment to the date of the first documented event of tumor progression or death in the absence of disease progression. Overall survival was defined as the time from the first day after the last radio-chemotherapy treatment until death from any cause.

Relevant clinical characteristics included age, gender, smoking status, tumor histology, ECOG performance status, as well as the blood values of albumin and CRP.

### 2.3. Study Oversight

The study protocol and amendments were approved by the ethics committee of Oldenburg University (2019-080), and the study was performed in accordance with the Declaration of Helsinki. Moreover, the names of the patients and all other confidential information are subject to medical confidentiality and the provisions of the Federal Data Protection Act (GDPR). Sample data were only passed on in anonymized form. Third parties did not have access to the original documents.

### 2.4. Statistical Analysis

The statistical analysis was carried out using the Kaplan–Meier method and the log rank test. A *p* value of 0.05 was set as the significance threshold for each statistical test. Furthermore, based on the survival curves of the PACIFIC study, we performed a power analysis. Therefore, we used the survival rates of both groups with a follow-up time of 45 months, considering that differences in the PACIFIC study were first noticed after 15 months. The power of the study, with a follow-up period of at least 21 months, was 0.83.

Furthermore, we constructed Cox regression models to predict the median OS separately in multivariate analyses. In the Cox regression models, we used the following covariates: age at diagnosis, sex, histology, smoking status, ECOG, albumin, CRP, and PD-L1 status. Subsequently, the covariates were excluded step by step with the reverse procedure using the random forest method. The results are presented as hazard ratios (HR) with 95% confidence intervals and *p* values.

## 3. Results

### 3.1. Patients

Table 2 shows the patient’s characteristics. A total of 99 patients with stage III NSCLC were included in the retrospective cohort study. The median age of the patients was 61 years, with a range between 41 and 82 years. The gender distribution was 64.6% (n = 64) men and 35.4% (n = 35) women. With respect to smoking status, patients were mainly former heavy smokers (48.5%, n = 48) and active smokers (38.4%, n = 38). Most of the patients had an ECOG of 0 or 1 (78.8%; n = 78). The distribution of non-squamous and squamous carcinomas was similar to the PACIFIC trial, with almost 1:1 squamous vs. non-squamous carcinomas. All chemotherapy was platinum-based; most frequently, a combination of cisplatin and vinorelbine was used (58.7%, n = 58), followed by cisplatin monotherapy (20.2%, n = 20). The distribution of PD-L1 values was 34.3% < 1 (n = 34), 65.7% ≥ 1 (n = 65), 70.7% < 50% (n = 70), and 29.3% ≥ 50% (n = 29). The median proportion of PD-L1-expressing immune cells was 5%.

### 3.2. PFS

A longer PFS is significantly associated (*p* = <0.001) with a PD-L1 expression ≥ 50% (n = 29). For patients with a PD-L1 expression ≥ 50% (n = 29), the PFS was 19 months (CI 2.1–35.9); for patients with a PD-L1 expression <50% (n = 70), the PFS was 17.9 months (CI 8.1–11.9). In the IC proportion scale (<5%, <10%, ≥10%) and radiation dose, no significant difference was seen.

Figure 1 shows the PFS of the retrospective cohort according to the PD-L1 status. A total of 99 cases were included in this evaluation.

The median PFS of patients with a PD-L1 expression of <1% (34.3%, n = 34) was 19 months (CI 6.4–11.6) vs. 12 months (CI 9.8–14.2) for patients with a PD-L1 expression of ≥1% (65.7%, n = 65). The log rank test showed no statistically significant difference, with a *p* value of 0.112. The tail of the PFS curve was higher for patients with a PD-L1 expression > 1% vs. <1%.

### 3.3. OS

A longer OS is significantly associated (*p* = 0.016) with a radiation dose of ≥60 Gy (n = 63). For patients who received a radiation dose of ≥60 Gy (n = 63), the OS was 30.3 months (CI 23.9–36.7); for patients who received a radiation dose of <60 Gy (n = 36), the OS was 17.9 months (CI 12.4–23.6). On the IC proportion scale (<5%, <10%, ≥10%), no significant difference was seen.

Figure 2 shows the survival curve of 99 NSCLC patients according to their PD-L1 status. The median OS of patients with a PD-L1 expression of <1% (n = 34) was 20 months (CI 10.5–29.5) and 27.9 months (CI 16.5–39.3) for those with a PD-L1 expression of ≥1% (n = 65). The log rank test showed no statistically significant difference, with a *p* value of 0.734.

### 3.4. Blood Parameters

A CRP value was determined pre-therapeutically in 79 patients. The distribution was as follows: 22 patients had values below 5.61 mg/L; in 9 patients, the value was between 5.61 and 8.58 mg/L; and 48 patients had a value of >8.58 mg/L. The median survival time for CRP values of <5.61 mg/L was 30 months (CI 23.9–36.5), and for values above 5.61 mg/L, it was 14 months (CI 7.7–21.1). Although there were numerical differences, these were not statistically significant in the log rank test, with a *p* value of 0.098 (Figure 3).

Of the 99 patients, the albumin value was determined in 64 patients before the first radio-chemotherapy treatment. The distribution was 65.6% (n = 37) < 35 g/L and 34.4% (n = 26) ≥ 35 g/L. The median overall survival in the group with an albumin value of <35 g/L was 28 months (CI 16.8–39.3). In the group with a value of ≥35 g/L, the median overall survival was 23.9 months (CI 10.3–37.4). The log rank test showed no statistically significant difference, with a *p* value of 0.868. Figure A1 (Appendix A) shows the OS as a function of the albumin values. Most of the patients had a quotient of >0.22 (n = 40), followed by a quotient of <0.14 (n = 22). The longest median survival was 27.8 months (CI 14.7–40.9) for those who had a quotient of <0.22. In the group of patients with a quotient of ≥0.22, the median survival time was 23.9 months (CI 6.3–41.6). The log rank test showed no statistically significant difference, with a *p* value of 0.759 (Figure A2 Appendix A).

### 3.5. Cox-Regression

In a multivariate analysis of the endpoint OS, the following covariates were included: *Age, sex, histology, smoking status, ECOG, radiation dose, and PD-L1 status.* The model with the seven covariates was statistically significant (likelihood = 499.328; Chi-square = 16.127; *p* = 0.03; Table 3). *Radiation dose* and *age at diagnosis* contributed significantly to the survival time model and were independent factors. Figure 4 and Figure 5 show the hazard curves of the covariates *radiation dose* and *age at diagnosis*.

## 4. Discussion

In the current retrospective analysis, we tried to simulate the control group of the PACIFIC study and test for differences in the OS relative to PD-L1 expression. No differences in the OS between PD-L1 <1% and ≥1% were found.

The population enrolled in the PACIFIC trial and the retrospective population analyzed in the current study were very similar. In the PACIFIC study, 70.1% (n = 500) of the patients were male and 29.9% (n = 213) were female [9]. In our cohort, the ratio was similar, with 64.6% (n = 64) male and 35.4% (n = 35) female patients.

Also, no differences were seen in the median age, which was 64 years in the PACIFIC study and 61 years in this retrospective analysis. Furthermore, the histologies were similar, with 45.7% (n = 326) of the patients having squamous cell carcinomas and 54.3% (n = 387) having non-squamous carcinomas, mainly adenocarcinomas, in the PACIFIC trial, and 48.5% (n = 48) having squamous carcinomas and 51.5% (n = 51) non-squamous cell carcinomas in our retrospective cohort.

In the PACIFIC trial, most patients in the control arm had an ECOG 1 (51.45%), as in our retrospective study (45.5%, n = 45).

The distribution of PD-L1 values in the PACIFIC study was 67.2% (n = 303) ≥ 1% and 32.8% (n = 148) < 1%. In our retrospective cohort, the distribution of PD-L1 expression values was similar, with 65.7% (n = 65) ≥ 1% and 34.3% (n = 34) < 1%. These results are also in concordance with the distribution of PD-L1 expression values < 1% from 30 to 35% and PD-L1 values ≥ 1% from 65 to 70% found in the literature as well as in a German prospective cohort, CRISP [16,17,18,19]. This results in a high level of consistency in PD-L1 expression in the patient groups examined, which also speaks for the stability of the detection method for PD-L1 expression [11,12,13]. The methodology used to evaluate PD-L1 expression was identical to the one used in the PACIFIC study.

### 4.1. Differences in Populations

The number of patients in whom the PD-L1 status was available was higher in the PACIFIC study (451) as compared to our analysis (99). In the PACIFIC study, smoking status was divided into current smoker (16.4%, n = 117), former smoker (74.6%, n = 74.6), and never smoker (9%, n = 64). In our retrospective cohort, significantly more patients were current smokers (38.4%, n = 38). A total of 48.5% (n = 48) of them had previously smoked and only 4% (n = 4) never smoked.

### 4.2. Comparison of PFS and OS as a Function of PD-L1 Expression and Blood-Based Biomarkers

The PACIFIC study included patients regardless of PD-L1 expression who did not progress after radio-chemotherapy with a platinum combination therapy and a dose of at least 54 Gy and who did not have radiation pneumonitis. A significant number of patients in the PACIFIC study received carboplatin (301 pts), while cisplatin combinations were administered in 395 pts. In our cohort, except for nine patients receiving carboplatin, all patients were treated with cisplatin; however, 20% only received cisplatin monotherapy. Inclusion of the patients in the PACIFIC trial was not stratified for PD-L1 expression; PD-L1 expression was studied in a subpopulation of patients that had given optional archival tissue specimens for analysis. The PACIFIC study had two co-primary endpoints, PFS and OS. Both were highly positive for the total patient population. The study is important for the patients with NSCLC, because it showed for the first time that in the curative situation, an immune checkpoint inhibitor improved survival with a statistically highly significant increase. The last 5-year analysis showed an OS improvement of 29% in the overall cohort and 40% in the PD-L1 ≥1% cohort. Post hoc, the approval authorities required the analysis of patients with PD-L1 expression <1% and ≥1%. For this purpose, the patients were subsequently examined for PD-L1 expression, which was successful in 63% of the patients. In this so-called biomarker group, as in the ITT population, a statistically significant PFS and OS advantage with durvalumab was shown. The subgroup analysis of the patients with a PD-L1 expression of ≥1% showed an advantage for durvalumab that was highly statistically significant. An analysis of the population with PD-L1 expression <1% showed that no OS benefit was achieved with durvalumab. Strikingly, the OS was also very high in the control group (i.e., the group without durvalumab), similar to that in the PD-L1 ≥ 1% group that had received durvalumab, and significantly higher than the OS in the group with a PD-L1 expression ≥ 1% treated with placebo (control group). Hence, it may be considered that after radio-chemotherapy, an initial PD-L1 expression of <1% has a favorable prognostic value.

In our retrospective analysis, all patients were tested for PD-L1. The survival time analysis showed that the PD-L1 status has no significant effect on the OS in the largest ever studied cohort of patients treated with definitive radio-chemotherapy for stage III NSCLC. There was also no significant difference when assessing the influence of PD-L1 expression on PFS. Thus, the hypothesis that the lack of PD-L1 expression is associated with a favorable prognosis after radio-chemotherapy could not be confirmed in the present patient collective.

These results are consistent with two other studies that have retrospectively studied the impact of PD-L1 expression on PFS and OS after radio-chemotherapy in two smaller cohorts and one larger cohort. In the studies by Tokito [12], Tufman [13], and Vrankar et al. [14], no differences were seen in the PFS and OS regarding PD-L1 expression in 74 pts (Tokito), 78 pts (Tufman), and 117 pts (Vrankar). In another study by Gennen including only 31 patients, a statistically significant difference was found; however, the number of patients was very small, suggesting an accidental finding [11]. Interestingly, the patients with a PD-L1 expression of ≥50% in our study had a superior PFS and OS than the patients with a PD-L1 expression of <50%, suggesting a positive prognostic impact of a high PD-L1 expression independently of ICI treatment. However, this finding is most likely as accidental as the finding of a positive impact on OS correlated with a PD-L1 expression of <1% in the PACIFIC trial.

In the PACIFIC study, the PD-L1 analysis was a post hoc analysis, i.e., the patients were not stratified based on this analysis, so there may have been imbalances in the two groups. However, based on this post hoc analysis, the EMA has limited the approval for durvalumab to patients with a PD-L1 expression of ≥1%, which has been scientifically criticized because of the post hoc analysis. Our data, together with the published data, show that PD-L1 expression is most likely not prognostic; therefore, this suggests that the lack of difference in the PD-L1 <1% group in the PACIFIC trial might be an accidental finding.

A total of 387 NCSLC patients of all stages were included in the “post-diagnostic C-reactive protein and albumin predict survival in Chinese patients with non-small-cell lung cancer: a prospective cohort study” [15]. The gender distribution, with 62.3% male and 37.7% female patients, was similar to our cohort (65.3% male, 34.7% female). In their study [15], it was found that an increased CRP value (>8.58 mg/L), a decreased albumin value (<35 g/L), and a CRP/albumin quotient of >0.22 had a negative prognostic impact for OS and PFS. However, no statistically significant impact on OS was seen in the non-metastatic setting. However, in other studies, a prognostic impact of CRP and albumin was also shown in the non-metastatic setting [20,21,22,23].

In our cohort, we could show that patients with a CRP value > 8.58 mg/L had the lowest probability of survival, while CRP values < 5.61 mg/L are associated with a longer OS; however, these differences were not statistically significant, presumably due to the small number of cases. Thus, our patient collective supports the data of the Chinese study to the extent that a numerically lower OS was found in patients with stage III NSCLC with high CRP values.

### 4.3. Strengths and Weaknesses of Data Analysis

All patients who were treated at the Pius Hospital between 2011 and 2018 and who met the inclusion criteria were included in this retrospective study. Staging was uniformly performed according to the recommendations of the S3 guidelines and the Oncopedia guidelines in place in Germany, i.e., all patients had to undergo a PET-CT and an MRI to exclude distant metastases, indicating that the staging in our cohort was even more precise than the staging in the study by Vrankar et al. [14] as well as in the PACIFIC trial. Patient selection for definitive radio-chemotherapy was performed throughout the years by an identical interdisciplinary team of pneumologists, radiation oncologists, oncologists, thoracic surgeons, nuclear medicine specialists, and pathologists, suggesting a high consistency of treatment and uniformness of the patient cohort. The data collection was almost complete, standardized and of high quality, so that very reliable statements about the patient characteristics could be obtained. The determination of the PD-L1 status was performed in a ring-certified pathology laboratory with the identical technique used in the PACIFIC study. The agreement with the distribution of PD-L1 values in the literature also means that the test is accurate and meaningful. In addition, the patient collective is very similar to that of the PACIFIC study.

A weakness of the data analysis is the small study population compared to the prospective randomized trial. In addition, not all clinical characteristics could be evaluated in a few patients because not all data had been collected. This was particularly difficult when evaluating the smoking status. In nine out of ninety-nine cases, no smoking status was documented. The same was true for blood values of albumin and CRP that were not available in all patients within a period of 14 days before the start of chemoradiotherapy.

## 5. Conclusions

The aim of the study was to test the assumption that in patients with stage III NSCLC after receiving definitive radio-chemotherapy, PD-L1 expression had a prognostic impact on OS. For this purpose, a retrospective analysis of the largest patient group published yet, uniformly treated in stage III with radio-chemotherapy and in whom PD-L1 expression was known, with respect to OS and PFS, was performed. Furthermore, albumin and CRP values and the ratio of the two biomarkers were also analyzed.

We found no impact of PD-L1 expression on PFS or OS in this retrospective cohort. Furthermore, a high CRP value seems to be associated with a worse OS. However, this finding is not statistically significant, most likely due to the small number of patients. A larger analysis using other patient data sources could further validate our findings.

In the future, based on the findings of the PACIFIC study as well as our retrospective analysis, stratification on the basis of PD-L1 expression might be important for the interpretation of clinical trials using immune checkpoint inhibitors. Furthermore, a meta-analysis of all real-world data might be able to determine more precisely the prognostic impact of PD-L1 expression after radio-chemotherapy.

## Figures and Tables

**Figure 1 biomedicines-12-00688-f001:**
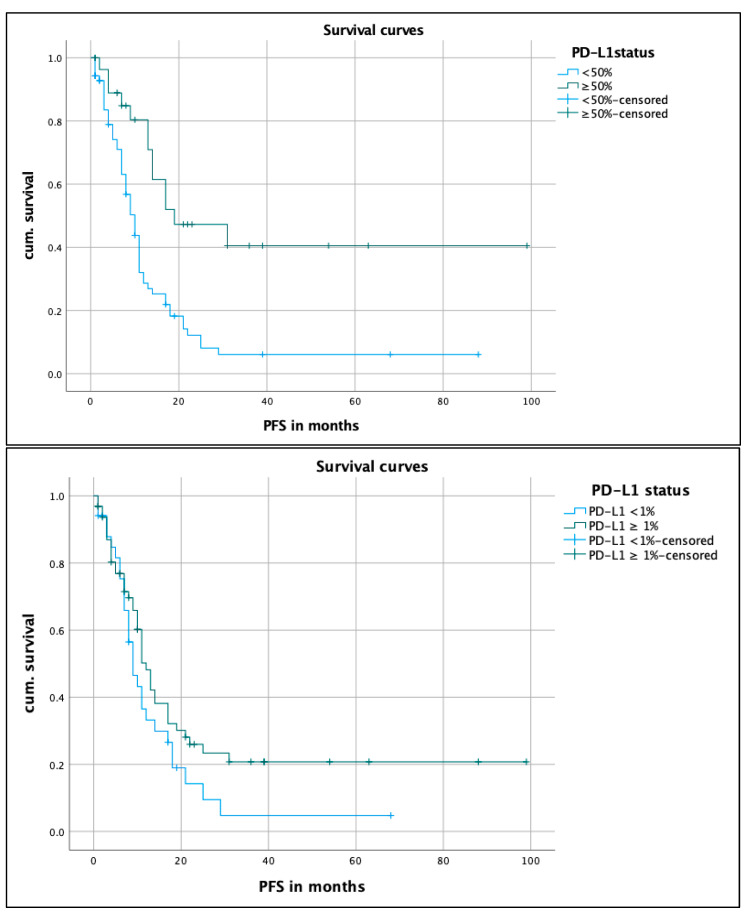
PFS of the patients depending on PD-L1 status. Kaplan–Meier curve on median PFS on patients with PD-L1 < 1% compared to patients with PD-L1 ≥1%; median PFS in months; for the calculation of the *p* value, the log rank test was used. PFS, progress-free survival; PD-L1, programmed death-ligand 1; cum., survival cumulative survival.

**Figure 2 biomedicines-12-00688-f002:**
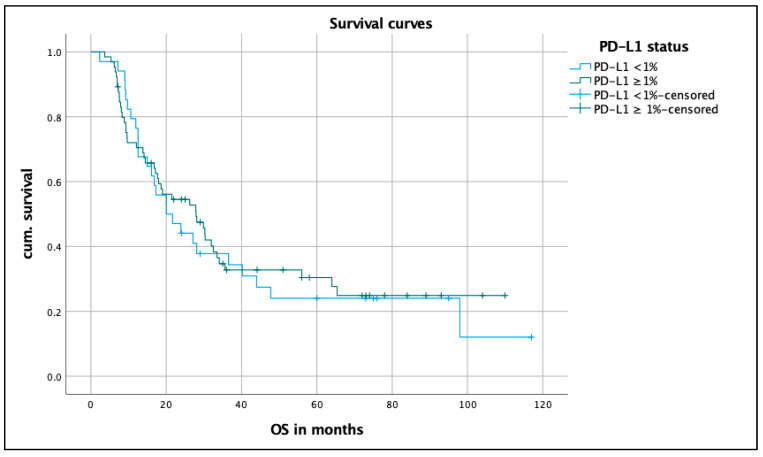
OS of the patients depending on PD-L1 status. Kaplan–Meier curve on median OS on patients with PD-L1 <1% compared to patients with PD-L1 ≥ 1%; median OS in months; for the calculation of the *p* value, the log rank test was used. OS, overall survival; PD-L1, programmed death-ligand 1; cum., survival cumulative survival.

**Figure 3 biomedicines-12-00688-f003:**
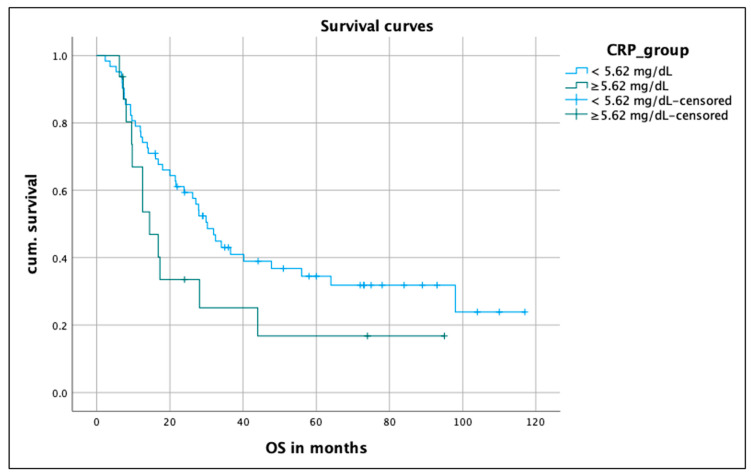
OS of the patients depending on CRP value. Kaplan–Meier curve on median OS on patients with PD-L1 < 1% compared to patients with PD-L1 ≥1%; median OS in months; for the calculation of the *p* value, the log rank test was used. OS, overall survival; PD-L1, programmed death-ligand 1; cum., survival cumulative survival.

**Figure 4 biomedicines-12-00688-f004:**
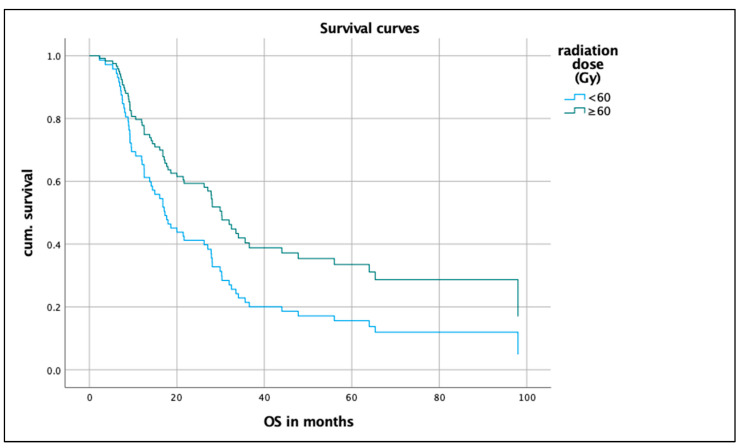
Hazard curve of radiation dose.

**Figure 5 biomedicines-12-00688-f005:**
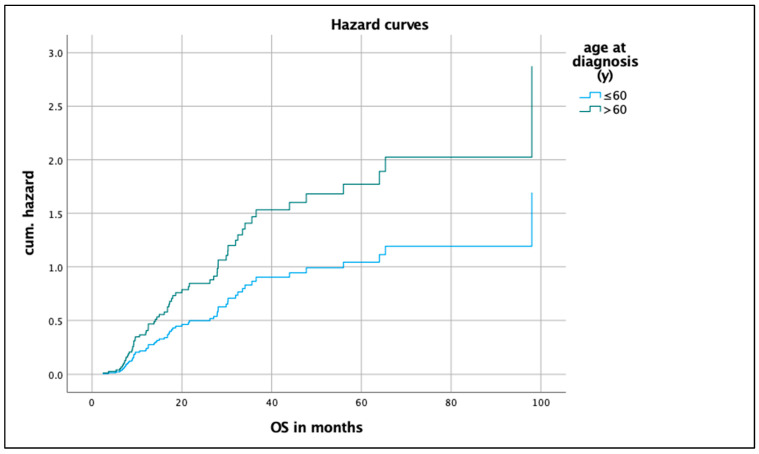
Hazard curve of age at diagnosis.

**Table 1 biomedicines-12-00688-t001:** Study results of the prognostic value of PD-L1 expression in NSCLC.

Study	Year	N	Results
A. Tufman et al. [13]	2021	78	Tumor PD-L1 expression did not correlate with PFS or OS.
T. Tokito et al. [12]	2016	74	PD-L1 expression < 1% associated with favorable OS.
K. Gennen et al. [11]	2020	31	PD-L1 expression < 1% associated with improved OS, PFS, and local control.
M. Vrankar et al. [14]	2020	117	Tumor PD-L1 expression did not correlate with PFS or OS.

**Table 2 biomedicines-12-00688-t002:** Baseline characteristics; stratification factors.

Variable	Total (n)	%
Patients	99	100
Observation period *		
Alive	32	32.3
Dead	67	67.7
Age (years), median (range)	61 (52.8–69.2)	
Sex		
Male	64	64.6
Female	35	35.4
Pack Years	40	
Smoking Status		
Light Smoker	2	2.0
Ex-Heavy Smoker	48	48.5
Current Smoker	38	38.4
Never Smoker	4	4.0
Unknown	7	7.1
Tumor histology type		
Non-squamous	51	51.5
Squamous	48	48.5
Chemotherapy		
Cisplatin	20	20.2
Cisplatin + Pemetrexed	7	7.1
Cisplatin + Vinorelbine	58	58.7
Cisplatin + Etoposid	2	2.0
Cisplatin + Paclitaxel	3	3.0
Carboplatin	9	9.1
ECOG status		
0	33	33.3
1	45	45.5
2	4	4.0
Missing	17	17.2
Mean Albumin	3.3	
Mean CRP	3.0	
Mean Gy	63.2	
PD-L1 status		
<1%	34	34.3
≥1%	65	65.7

* Observation period: 1 January 2011–15 March 2020; ECOG, Eastern Cooperative Oncology Groups; Gy, Grey; Programmed death-ligand 1.

**Table 3 biomedicines-12-00688-t003:** Multivariate analysis of overall survival of 99 lung cancer patients.

Multivariate Analysis				
OS				
Covariate	Category	HR	95%-CI	*p* Value
**Sex**	Male	0.293	0.758–2.371	0.132
	Female (RC)			
**Histology**	Non-squamous	0.179	**0.699–2.047**	0.514
	Squamous (RC)			
**Radiation dose (Gy)**	≥60	0.675	1.142–3.377	**0.015**
	<60 (RC)			
**Age at diagnosis (y)**	>60	0.497	0.953–2.832	0.074
	≤60 (RC)			
**ECOG**	0–1	0.411	0.749–3.035	0.250
	≥2 (RC)			
**Smoking status**	never/light	0.521	0.566–5.010	0.349
	current/ex heavy (RC)			
**PD-L1 (%)**	<50	0.276	0.742–2.339	0.346
	≥50			

RC—reference category.

## Data Availability

Data are contained within the article.

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
