# Peer review of "Evaluation of the Prognostic Impact of SP263-Evaluated PD-L1 Expression in Patients with Stage III Non-Small Cell Lung Cancer (NSLC) Treated with Radio-Chemotherapy"

_biomedicines, 2024, doi:10.3390/biomedicines12030688_

Round 1

Reviewer 1 Report

Comments and Suggestions for Authors

Stage III NSCLC is the most clinically complex stage of lung cancer due to its heterogeneity. The manuscript has value to the clinical practice and patients.

  1. 1. Materials and Methods: It would be good to include a study algorithm for patient selection which would be beneficial for readers.
  2. 2. Results - Blood Parameters: It would be interesting to explore if other parameters, such as the derived neutrophil-to-lymphocyte ratio (dNLR), have predictive value.
  3. 3. Table 1: The percentage of "unknown" in smoking status appears incorrect. Please correct.
  4. 4. Discussion: Discuss more about the toxicity profile in your cohort.
  5. 5. Discussion: The study could benefit from a broader review of existing research and more real-world studies. Adding a table summarizing the analyses of a few studies could be helpful.
  6. 6. Ethical Considerations: Ensure all ethical guidelines are rigorously followed and documented.
  7. 7. Future Directions: Provide more explicit and detailed suggestions for future research based on the findings.
  8. 8. References for Authors: PMID 33312885
Comments on the Quality of English Language

Minor improvement 

Author Response

Dear Colleague,

Thank you very much for your review and the helpful suggestions.

1) The study was conducted as a complete survey of real-world data from routine clinical care. All patients who met the inclusion criteria were included. We have explained this in more detail in the text.

2) We did not have sufficient blood values in all patients to generate meaningful results on the neutrophil-to-lymphocyte ratio (dNLR) or other blood values. Therefore, we looked at CRP and albumin.

3) We have corrected this.

4) We did not describe and further investigate the toxicity profile in detail, as the patients were all treated simultaneously with radiochemotherapy. The known toxicities occurred under RCT and reproducing these would not add to the readers knowledge and was therefore not the aim of our study.

5) For the sake of clarity, we have presented the few studies available on this topic in a table. 

6) As described in section 2 Materials and Methods under Study Oversight, we are pleased to confirm that we have adhered to and met all ethical standards.

7) We have clarified this in the text. 

Reviewer 2 Report

Comments and Suggestions for Authors

In general, this manuscript on "Evaluation of the prognostic impact of SP263-evaluated PD-L1 expression in patients with non-small cell lung cancer (NSLC) stage III treated with radio-chemotherapy" is well-written regarding scientific aspects and format.

My comments:

1. The PDL1<1% subgroup has 34 patients and the PDL1>1% subgroup has 65. The number of patients are quite imbalanced in the subgroups. This can affect the conclusion and should be given in the weaknesses of the sudy.

What was the calculated sample size for each subgroup (PD-L1<1% and PD-L1>1%). Could the authors reach these sample sizes?

2. The authors should investigate whether there is a correlation between PD-L1 and albumin levels, and PD-L1 and CRP levels.

Author Response

Dear colleague, thank you very much for your review and your questions, which we are happy to answer.

1. the study was planned as a full survey of real world data from daily routine care. We looked at all patients who were treated by us during this period. We clarified this in the text.

Otherwise, a distribution of approx. 30% per group is given in the literature. Approx. 30% in the group with a PD-L1 expression <1%, 30% in the group from 1-49% and 30% in the group over 50%. Since we divided these into 2 groups, the study population is representative, even if you look at the exact PD-L1 distribution.

2. we have recalculated the data as requested, thank you for pointing this out. The distribution shows no significant differences in the cross-tabulations and Chi2 tests, neither for CRP nor for albumin. We have added this information to the text. For example, a CRP value of <5.61 with a PD-L1 expression of <1% was present at 40.9% (PDL1>1% 59%) and >5.61 at 54.% (PDL1>1^% 58.8%)

Round 2

Reviewer 1 Report

Comments and Suggestions for Authors

Thanks for your update. 

Comments on the Quality of English Language

minor improvement. 

Reviewer 2 Report

Comments and Suggestions for Authors

The authors have revised the manuscript according to my comments.